# Estimation of Rice Biomass at Different Growth Stages by Using Fractal Dimension in Image Processing

**Yijun Hu, Jingfang Shen \* and Yonghao Qi**

College of Sciences, Huazhong Agricultural University, Wuhan 430070, China;
huyijun@webmail.hzau.edu.cn (Y.H.); 1780856632@webmail.hzau.edu.cn (Y.Q.)
\* Correspondence: sjf_712@mail.hzau.edu.cn

**Abstract:** Rice has long served as the staple food in Asia, and the cultivation of high-yield rice crops draws increasing attention from academic researchers. The prediction of rice growth condition by image features realizes nondestructive prediction and it has great implications for smart agriculture. We found a special image parameter called the fractal dimension that can improve the effect of the prediction model. As an important geometric feature, the fractal dimension could be calculated from the image, but it is rarely used in the field of rice growth prediction. In this paper, we attempt to combine the fractal dimension with traditional rice image features to improve the effect of the model. The thresholding method is used to transform the cropped rice image into binary image, and the box-counting method is used to calculate the fractal dimension of the image. The correlation coefficients are calculated to select the characteristics with a strong correlation with biomass. The prediction models of dry weight, fresh weight and plant height of rice are established by using random forest, support vector regression and linear regression. By evaluating the prediction effect of the model, it can be concluded that the fractal dimension can improve the prediction effect of the model. Among the models obtained by the three methods, the multiple linear regression model has the best comprehensive effect, with the dry weight prediction model $R^2$ reaching 0.8697, the fresh weight prediction model $R^2$ reaching 0.8631 and the plant height prediction model $R^2$ reaching 0.9196. The model established in this paper has a fine effect and has a certain guiding significance in rice research.

**Keywords:** rice biomass; fractal; machine learning; predictive model

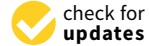

## 1. Introduction

Agriculture is the foundation of social stability and national development in many countries, especially in developing countries. Adequate supply of grain and other basic agricultural products is an irreplaceable foundation for ensuring sufficient stability of market prices. Meanwhile, food security is a crucial support for national economic security. Therefore, it is a particularly urgent and meaningful task to strengthen the macro-control of the food market and ensure food security and market stability at present.

Rice plays an essential role in agricultural production as the most significant food crop [1]. With the rapid increase in the population, the country's demand for food is also growing. We have problems with declining soil quality and eutrophication [2]. Since rice plays a key role in food security, it is necessary to study the growth status of rice. Industrialization reduces the arable land area, so it is required to increase the yield of rice in order to increase the total yield of rice. Therefore, rice growth and cultivation of high-yield varieties has become a major research issue in agricultural research institutions [3]. The study of rice growth is very critical to political, economic and social stability.

Due to the development of computer information technology, image-processing technology is becoming more and more advanced. In order to detect and extract data from the overall growth state of rice, phenotypic feature data extraction from rice image has been widely used. Compared with artificial experiments, this method can reduce subjective

errors and improve the accuracy of data. The main content of image processing is to cut the areas in the original image that need to be studied, and then extract the features of specific areas according to color components and texture features [4].

Fractal geometry has a self-similar character. When the geometries with fractal characteristics are magnified at different multiples, the observed properties are similar and have a fractal dimension in space [5,6]. The concept of fractals was first put forward by B. B. Madelbrot in 1973. He defined a fractal as a set, which expresses the symmetry or self-similarity of the whole and the part in a certain sense. There are many fractal phenomena in nature, such as coastlines, lightning, human lungs, material surfaces, etc., which can be approximately regarded as fractal sets [7].

However, the typical fractal sets are specially constructed by mathematicians, so they have a standard and strict self-similarity. However, in nature, most of the objects we study do not have strict self-similarity characteristics, but only meet statistical self-similarity. Therefore, in research, we mainly carry out approximate processing on the objects to explore the quasi-self-similarity [8].

The dimension of fractal geometry is usually not an integer dimension, so the integer dimension used in the traditional Euclidean space cannot describe the fractal shape. Mandelbrot proposed the concept of a non-integer dimension or fractal dimension [9]. The fractal dimension describes a complex ratio that reflects how the details of the pattern change with the scale of the measurement [10]. It is used in all areas of science because it provides a measure of the complexity and irregularity of a given object [11]. Complexity is a change in detail and scale. As the fractal dimension increases, the complexity of the object also increases.

As a long-neglected part of geometry, fractals can help us to study nature from a new perspective and find order in apparent disorder [12]. Since the 1970s, the application of fractal theory has developed rapidly and gradually become an important new subject. Fractals have been widely used in natural and social sciences such as biology, chemistry, physics, material science, computer graphics, seismology, economics and so on. It is now one of the frontier research disciplines of many disciplines all over the world [13]. It provides a new method for researchers to solve traditional nonlinear problems more accurately.

The fractal dimension has been widely used in various fields. Pinavega Rogelio et al. [14] proposed an automatic prediction method for sudden cardiac death (SCD) based on a fractal dimension algorithm and a fuzzy logic system. In the paper, five kinds of fractal dimensions were used for experimental research. The results show that the method of basic fractal dimension could predict SCD events, and the prediction time was up to 60 min before the onset, with an accuracy of 91.54%. Cheng Liu et al. [15] proposes an improved DBC (IMDBC) to estimate the fractal dimension of three-dimensional (3D) pavement texture images based on a grid displacement mechanism. Combined with the contact characteristics of the road surface and the tire, the most suitable road surface with fractal texture was determined by the fractal stratification method. Compared with traditional DBC, the fitting accuracy of IMDBC is improved by 18.8% (full texture) and 900% (partial texture), respectively. Lucas Glaucio da Silva et al. [16] analyzed all the images using the FracLAC algorithm in the ImageJ computing environment to obtain the box fractal dimension results. They found that computer-aided diagnostic algorithms can benefit from box fractal dimension data; the cutoff value of the fractal dimension of the specific box produced 0~99% specificity in the diagnosis of breast cancer. Eloy Roura et al. [17] aimed to evaluate longitudinal changes in brain fractal geometry and its predictive value for disease progression in patients with multiple sclerosis (MS). The box number method was used to calculate the dimensionality of brain differentiation and the space between brain regions. Fractal geometric analysis of brain MRI found that patients had an increased risk of disability over the next 5 years. Shanshan Jin et al. [18] used pore size distribution curves and pore volume histograms to qualitatively analyze the pore structure before and after freeze-thaw. A fractal model was used to characterize pore distribution. A micro freeze-thaw damage model with fractal dimension as an independent variable was

established, and the relationship between the damage parameters calculated by the model and durability factors was analyzed. Parikshaa Gupta et al. [19] aimed to evaluate the value of fractal dimensions in differentiating benign and malignant HCGS endometrium in liquid cervical specimens. They suggest that fractal dimension analysis is an effective tool to distinguish between different types of cell groups. It is concluded that the fractal dimension detection of cervical cell carcinoma has high sensitivity and can be used as an effective screening method for differentiating benign and malignant cervical cell carcinoma. When the object is relatively complex, it can be studied from the perspective of fractals. The fractal dimension can reflect the complexity of the object, which may improve the analysis effect.

The traditional method of yield estimation is field sampling survey. Observers estimate the yield of a large area according to the growth condition of samples through observation sampling evaluation. This estimation method cannot be standardized and requires a large amount of manpower and material resources. At present, rice yield prediction is mainly divided into meteorological model prediction, remote sensing model prediction and image feature model prediction.

Forecasting based on a meteorological model is mainly to analyze the correlation between meteorological factors and yield. The key meteorological factors are selected to establish a model for yield prediction [20]. In order to explore the influence of meteorological factors on rice yield, Li Hongyan et al. [21] used meteorological data and rice yield data of Tongxiang city over 13 years. They used an exponential smoothing method to calculate the trend yield of rice, and conducted correlation analysis with the monthly average temperature, maximum temperature, minimum temperature, sunshine hours and precipitation in the rice growth period. The key factors affecting the yield of rice were determined, and the regression equation was established and tested. The average accuracy of the prediction model was up to 96.2%. Zuo Huiting et al. [22] studied climate change in different climatic zones, different climatic conditions and recent years. Combined with the variation trend of rice yield in different climatic zones, they analyzed the correlation between rice yield and climatic factors, and selected the main controlling factors of rice yield in each climatic zone. The regression model was established to forecast the production of different climate zones in the next five years, and the results are relatively stable. Meteorological models need a lot of meteorological data and yield data, and need to maintain the consistency of growth conditions. Therefore, the established models have poor generality and are difficult to be popularized. The prediction based on remote sensing and spectral model prediction is mainly to obtain the plant spectral index through a multi-spectral camera, select key factors and establish a prediction model based on yield data [23]. Wang Di [24] extracted vegetation index, end element abundance, texture features and other information by using ground hyperspectral data and multi-spectral data of the UAV platform. Rice yield estimation was studied by stepwise linear regression, BP neural network and random forest algorithm. Liu Shanshan et al. [25] obtained the normalized vegetation index (NDVI) of remote sensing data according to time series and evaluated it with Pearson product moment correlation coefficient (Pearson) of average rice yield in the field by comparing the mean value of NDVI combination in different time periods. NDVI data were used to establish several prediction models with rice yield, and the best model was selected. Remote sensing yield estimation is generally applicable to large-area yield estimation, but when the planting area is not large enough, the accuracy is often reduced, and it is difficult to obtain remote sensing data.

The prediction is made based on the image feature model, mainly through segmentation of RGB images, extraction of features and establishment of regression model [26]. Gong Hongju and Ji Changying [27] made a preliminary study on the relationship between texture features of wheat spike head image and yield by using MATLAB image-processing technology. The mathematical model of spike head image texture and yield was established by using multiple linear regression method, and 84.42% of the samples with an accuracy of more than 15% were measured by using the established model. Li Yinian et al. [28]

performed the segmentation of wheat ears through color space conversion and image-processing technology to identify the number of ears. By predicting the number of grains by panicle area and combining this with 1000-grain weight, they built a model to predict the wheat yield per unit area with an average accuracy of more than 90%. The image feature model can not only make predictions with damage, but also make predictions without damage, and the effect is better when the planting area is small.

Fractal dimension is an important image feature, but it is seldom used in rice yield estimation. The purpose of this study is to combine the characteristics of rice in the early growth stage with the fractal dimension; practical and low-cost models are established. These models can be used to predict the fresh weight, dry weight and plant height of rice, which need to be measured by machine and are highly correlated with yield. The model can not only provide a reference for countries to formulate food security strategies, but also help to measure the yield of smart agriculture, so this study has a strong practical value.

## 2. Materials and Methods

### 2.1. Data Collection

The rice test base of this experiment was located in the potting farm of Huazhong Agricultural University, and the rice used for the experiment was planted in plastic containers. The bottom diameter of the plastic container was 16 cm, the top diameter was 19.5 cm and the height of the container was 19 cm. Each container was loaded with 5 kg of air-dried soil, and rice seedlings were planted in it with the proper amount of water. A visible-light industrial camera (AVT Stingray FG504) was used to take pictures of rice. We obtained 424 RGB images in PNG format at a size of 2452 by 2056. This batch of data contains three growth stages, in which the first stage of rice is at the tillering stage, the second stage of rice is at the jointing stage and the third stage of rice is at the heading and grain-filling stage. The RGB images of rice obtained by camera are shown in Figure 1.

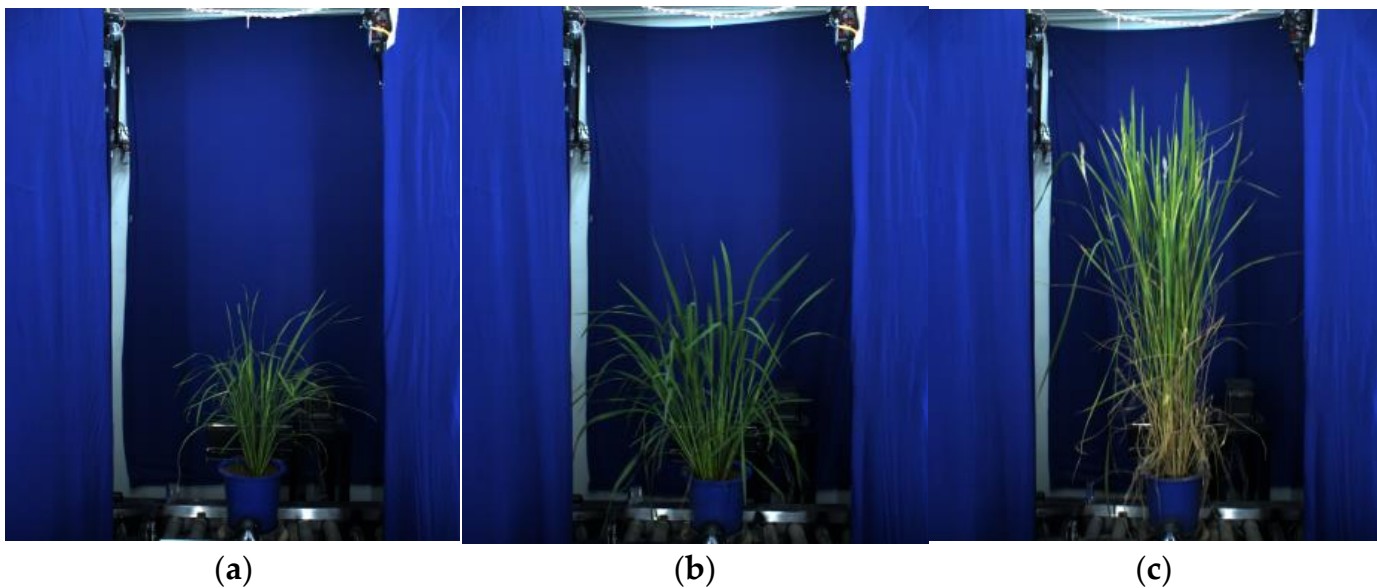

|    |    |    |
|:--:|:--:|:--:|
| (**a**) | (**b**) | (**c**) |

**Figure 1.** Sample RGB images of rice growth in three stages. (**a**) shows the tillering stage of rice; (**b**) shows the jointing stage of rice; (**c**) shows the heading and grain-filling stage of rice.

RGB images were transformed into binary images by the threshold method, and the formula was: 2G-R-B $\geq$ M (where G was the green component of the pixel, R was the red component and B was the blue component); M = 20 was the threshold value [29]. The binary images of the three batches of rice samples are shown in Figure 2.

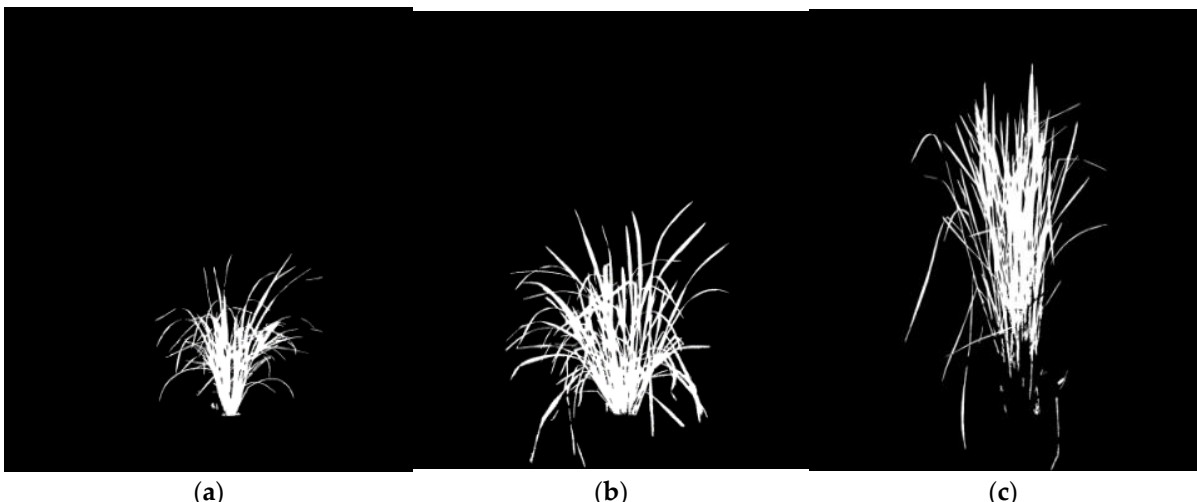

**Figure 2.** Sample binary graphs of rice growth in three stages. (**a**) shows the tillering stage of rice; (**b**) shows the jointing stage of rice; (**c**) shows the heading and grain-filling stage of rice.

The box-counting method is one of the most widely used methods to measure the fractal dimension. Its measuring principle is shown in Figure 3. The square grid with side length of $\delta$ was overlaid with the graph to be measured, and then the number of grids $N(\delta)$ that overlapped with the boundary curve was calculated. Then, we continuously reduced the side length of the grid, and we obtained the number of overlapping grids at different scales $\delta$. The relationship between the number of overlapped grids $N(\delta)$ and the grid side length $\delta$ satisfied Equation (1).

$$\log N(\delta) = D \log \frac{1}{\delta} + k \tag{1}$$

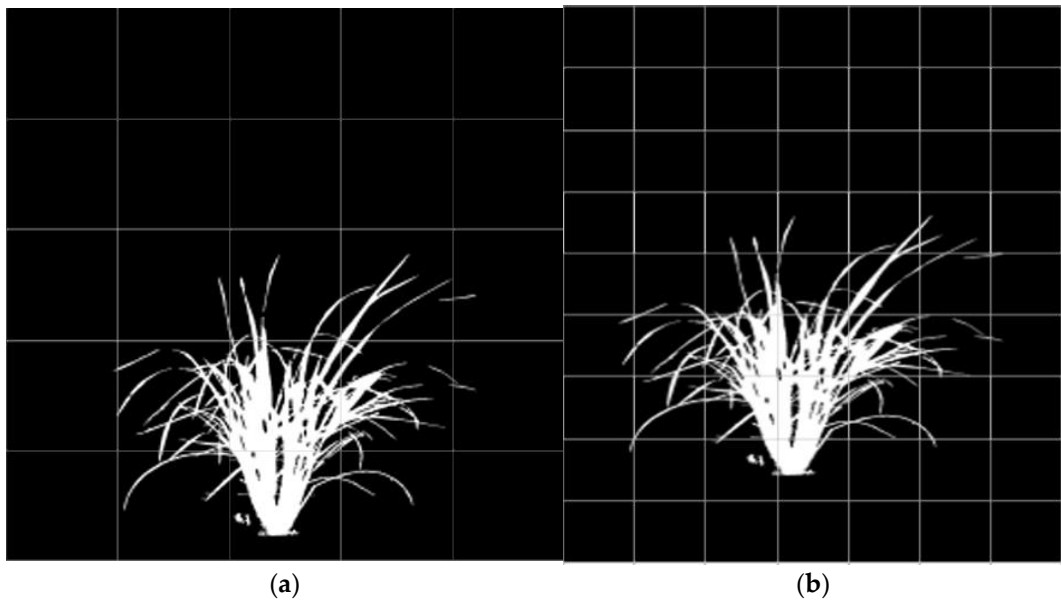

**Figure 3.** Box-counting method. (**a**) Cover the image with big boxes; (**b**) cover the image with small boxes.

In Equation (1), $k$ is a constant and $D$ is the fractal dimension. Then, the slope of the straight-line equation is the estimated value of the box-counting dimension.

Besides the fractal dimension, some traditional rice representations could be extracted based on the binary images of rice. By using the characterization extraction system, we

could obtain the characteristics of rice, including width, leaf area and texture characteristics, etc. Feature extraction was carried out once for each binary image. The characteristic variables and their abbreviations which were obtained from the images of rice are shown in Table 1.

**Table 1.** Characterization of rice and its corresponding symbols.

| Symbol | The Meaning of the Symbol |
|---|---|
| PW | Width of plant |
| PH(V) | Vertical height of plant |
| PH(V)/PW | Plant vertical height/width |
| PH | The height of rice leaves after straightening |
| PH/PW | Plant height/width |
| SA | Projection area of side view of rice plant |
| SA/PH(V) $\times$ PW | Projected area/multiply vertical height by width |
| IFD | Fractal dimension |
| SFD | Fractal dimension of the surrounding rectangle |
| G_g | Texture feature |
| f1–f12 | Relative frequency |
| LD1–LD6 | Structural parameters |

'PW' is the width of the rice plant. 'PH(V)' is the vertical height of the rice (the highest height of the rice in its natural state). 'PH' is the height of rice leaves after straightening and the process is simulated by image-processing method in the program. 'SA' is the lateral projection area of the rice plant. 'IFD' is the fractal dimension of the rice binary image obtained by box-counting method. 'SFD' is the box-counting fractal dimension based on the minimum bounding rectangle of the binary image. 'G_g' is a texture feature (gradient information), and we found that it could reflect the number of panicles in rice. 'f1–f12' are relative frequencies and 'LD1–LD6' are structural parameters. They all reflect the compactness of the plants. If their values are large, the plant is compact. In addition to the above rice characterization data, the fresh weight (g), dry weight (g) and plant height (cm) of rice could be obtained by manual measurement. Dry weight is the weight of dry matter, which can reflect the growth of the plant. Fresh weight can reflect the water content of the plant. Different rice varieties have different water content, and the direct difference between fresh weight and dry weight can reflect the water content information.

### 2.2. Select Feature Variable

In order to observe the relationship between the fractal dimension and the dry weight, fresh weight and plant height of rice, three point plots were drawn to reflect the distribution of the data. The three point plots are shown in Figures 4–6.

By observing these images, it can be seen that the fractal dimension is correlated with dry weight, fresh weight and plant height. In order to verify this, we need to calculate the correlation coefficient between them.

Pearson correlation coefficient can better reflect the degree of correlation between two variables, and its value is [−1, 1]. The correlation coefficient between characteristic variables and fresh weight, dry weight and plant height can be calculated.

Considering the correlation between explanatory variables and dependent variables, as well as the autocorrelation between explanatory variables, the correlation coefficients between the selected variables and their dependent variables are shown in Tables 2–4.

These variables are closely related to the dependent variables, so we can select these variables for modeling.

### 2.3. Modeling Methods

Three regression models are used for modeling and the effects of the models are compared.

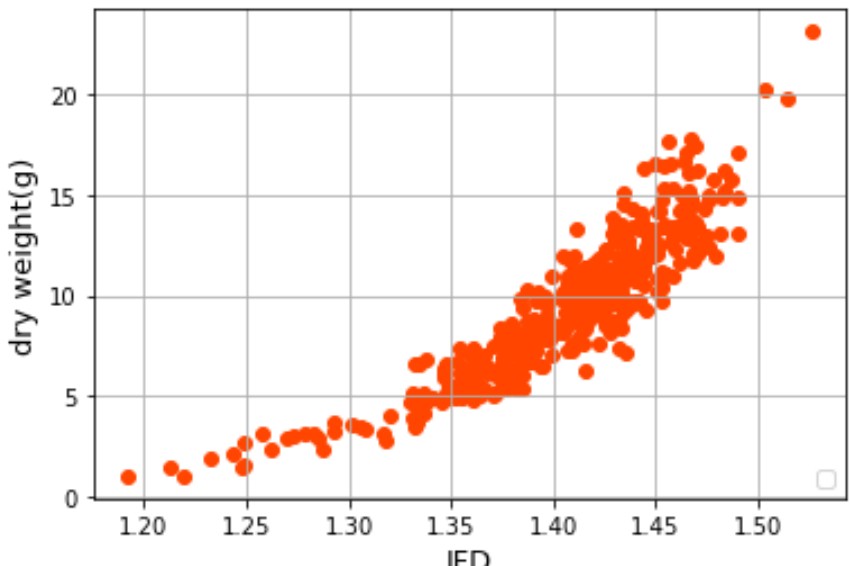

**Figure 4.** Scatter diagram of IFD and dry weight.

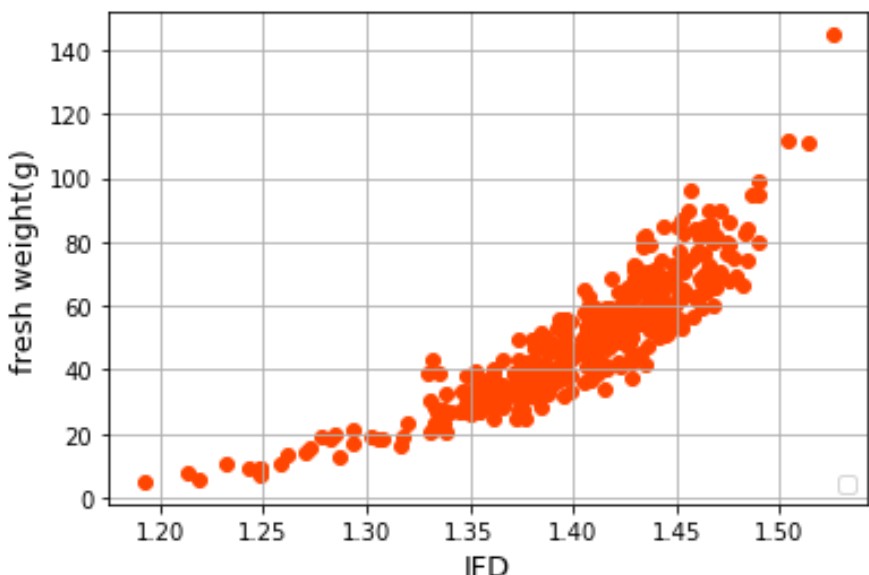

**Figure 5.** Scatter diagram of IFD and fresh weight.

**Table 2.** Correlation coefficient between characterization and fresh weight of rice.

| Variable | Correlation Coefficient with Fresh Weight |
|---|---|
| PW | 0.655757 |
| SA | 0.919291 |
| SA/PH(V) × PW | 0.530873 |
| IFD | 0.621371 |
| SFD | 0.885634 |
| G_g | 0.745343 |
| LD1 | −0.545686 |

**Table 3.** Correlation coefficient between characterization and dry weight of rice.

| Variable | Correlation Coefficient with Dry Weight |
|---|---|
| PW | 0.644520 |
| SA | 0.925310 |
| SA/PH(V) × PW | 0.525873 |
| IFD | 0.630591 |
| SFD | 0.893188 |
| G_g | 0.782164 |
| LD1 | −0.577148 |

**Table 4.** Correlation coefficient between rice characterization and plant height.

| Variable | Correlation Coefficient with Plant Height |
|---|---|
| PW | 0.600954 |
| PH(V) | 0.727988 |
| PH | 0.896098 |
| SFD | 0.696987 |
| G_g | 0.615098 |
| f1 | 0.559564 |
| LD2 | 0.521077 |

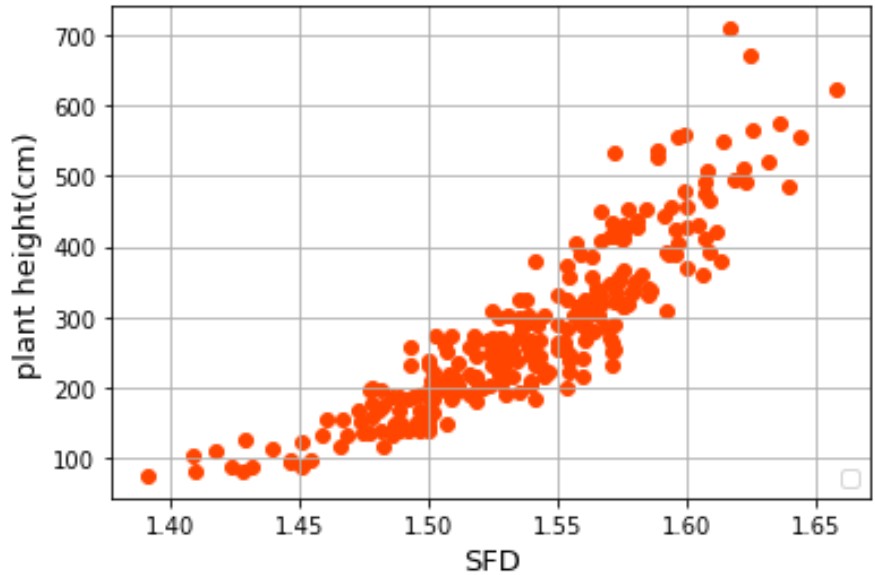

**Figure 6.** Scatter diagram of SFD and plant height.

### 2.3.1. Random Forest

Random forest is a kind of integration algorithm. Based on the bagging integration built with decision tree as the learning machine, random attribute selection is further introduced into the training process of the decision tree. Random forest is simple, easy to implement and has low computational overhead, but it can show powerful performance in many practical tasks, and is considered as a method to represent the technical level of ensemble learning [30]. Due to the universality of random forest, we try to use random forest to establish the biomass model of rice. The diversity of basic learners in random forests not only comes from sample perturbations, but also from attribute perturbations. Therefore, the generalization of the final integration can be improved by increasing the differences among individual learners. The initial performance of random forest is usually poor, but with the increase in basic learners, random forest can gradually converge to a lower generalization error and achieve better results. An important parameter in the

model, N_ESTIMATORS, is the number of trees in the forest; that is, the number of base learners. The influence of this parameter on the accuracy of the random forest model is monotonous. The larger N_ESTIMATORS is, the better the effect of the model will be. However, any model has a decision boundary. After N_ESTIMATORS reach a certain level, the accuracy of the random forest usually stops rising or starts to fluctuate. Moreover, the larger N_ESTIMATORS is, the more computation and memory is required, and the longer training time will be.

### 2.3.2. Support Vector Regression

Support vector regression is an extension of support vector machines. Assuming that we can tolerate a maximum deviation of $\varepsilon$, we will only calculate the loss if the absolute value of the difference between $f(x)$ and $y$ is greater than $\varepsilon$. It has a strong learning ability for small data sets, and can solve high-dimensional nonlinear problems by transforming them into linear ones through nonlinear transformation [31]. This batch of rice data contains more than 200 pieces of data, and the data volume is suitable for the use of support vector regression. Considering the application effect of each kernel function comprehensively, the Gaussian radial basis kernel function is finally determined to be used in the model established in this paper. The prediction model of support vector regression has two unknown parameters, penalty parameter C and kernel parameter Gamma. C represents the prediction ability of the model. The larger the C, the higher the degree of the model's learning from the samples, and the worse the prediction effect of the unknown data. On the contrary, the smaller C is, the higher the fault-tolerant rate of the model is, and the worse the fitting effect of the model is, but the prediction effect is relatively good. The kernel parameter gamma determines the distribution of the low-dimensional data mapped into the higher-dimensional space.

SVR model prediction is based on distance measurement and distance is very sensitive to different value ranges between features. Due to the different dimensionality of each feature of the data set, there is a large gap in the data size under different features, which will affect the results of data analysis. In order to eliminate the dimensionless influence among the indicators, data standardization is needed to solve the comparability among the data indicators. After the standardization of the original data, each index is in the same order of magnitude, which is suitable for comprehensive comparative evaluation. In this paper, normalization is adopted to process the data, and the formula is shown in Equation (2).

$$x' = \frac{x - \min(x)}{\max(x) - \min(x)} \tag{2}$$

Normalization can make the data map to the range of [0, 1], so as to eliminate the adverse effects caused by the strange sample data.

### 2.3.3. Linear Regression

Linear regression is a statistical analysis method that uses regression analysis in mathematical statistics to determine the interdependent quantitative relationship between two or more variables. It has a very wide range of applications. Since there is a linear relationship between the selected variables, multiple linear regression models can be used. The expression for this is $y = \omega^T X + e$. In the formula, $\omega^T$ is the regression coefficient matrix, and the number of regression coefficients in the matrix is equal to the number of independent variable X. Where e is the error, e is normally distributed and the mean is 0, which is $e \sim N(0, \sigma^2)$. In regression analysis, if there is only one independent variable and one dependent variable, and the relationship between the independent variable and the dependent variable can be approximated by a straight line, then this regression analysis is called unary linear regression. If there are two or more independent variables in the regression analysis, and there is a linear relationship between the dependent variable and the independent variable, then this regression analysis is called multiple linear regression. The effect of the linear regression model mainly depends on the data themselves, rather

than the ability to improve the model by adjusting parameters. As long as the linear connection of the data is strong, the least square method can be used to establish a linear regression model with fine effect.

### 2.4. Accuracy Evaluation

For model performance evaluation, $R^2$ and mean absolute percentage error ($MAPE$) are selected as the evaluation index.

The calculation for $R^2$ is shown in Equation (3).

$$R^2 = 1 - \frac{\sum\limits_{i=0}^{m} (y_i - \widehat{y}_i)^2}{\sum\limits_{i=0}^{m} (y_i - \widehat{y})^2} \tag{3}$$

where $y_i$ is the real result of the data, $\hat{y}_i$ is the predicted result of the model and $\overline{y}_i$ is the mean value of the data. In $R^2$, the numerator is the difference between the real value and the predicted value, which is the total amount of information not captured by the established model. The denominator is the amount of information carried by the real tag, so the numerator divided by the denominator measures the proportion of the amount of information not captured by the model to the amount of information carried by the real tag. Therefore, the closer $R^2$ is to 1, the better this model will be. Generally, when $R^2$ is greater than 0.6, the model can be considered to have a wonderful effect.

The calculation for $MAPE$ is shown in Equation (4).

$$MAPE = \frac{100\%}{m} \sum_{i=0}^{m} \left| \frac{\hat{y}_i - y_i}{y_i} \right| \tag{4}$$

$MAPE$ reflects the relative error of the model, and $MAPE = 0$ means that the model has no error. If $MAPE$ is greater than 100%, the model is too ineffective. The larger $MAPE$ is, the larger the error the model will have.

## 3. Results and Analysis

### 3.1. Random Forest Regression Model

According to the second chapter of this paper, the random forest regression model has a relatively important unknown parameter; that is, the number of regression trees in the forest, N_ESTIMATORS. In order to obtain the best effect of the model, we need to choose the size of the N_ESTIMATORS.

The size of the tree N_ESTIMATORS is selected by drawing the learning curve. By observing the variation of the model's $R^2$ with the N_ESTIMATORS, the overall size of the random forest is selected. We can see from Figure 7 that when the value of N_ESTIMATORS is greater than 50, $R^2$ tends to be basically stable. The larger N_ESTIMATORS is, the more computation and memory will be needed, and the longer training time will be. When N_ESTIMATORS is equal to 50, $R^2$ is in the front of the stable part and the calculation time is short. Therefore, the random forest model parameter in this paper is selected as 50.

Based on the data of the three growth stages of rice, the models of dry weight (g), fresh weight (g) and plant height (cm) of rice are established respectively with the characteristic variables selected above. The data is divided into training set and test set in a ratio of 4 to 1. In order to facilitate the comparison of model effects, we used the same test set and training set. The training set is used to build the model, and the test set is used to test the effect of the model.

After completing the establishment of the random forest regression model, the test set data are substituted into the model to obtain the corresponding prediction results. The predicted results are drawn into a line chart for fitting with the actual situation in the test set, as shown in the figure below. It can be seen from Figure 8 that the predicted results of the model are close to the actual situation, and the distribution of data is well

fitted. The prediction ability of the model is relatively stable. Under certain conditions, the predicted value of the random forest model established in this section has certain guiding significance.

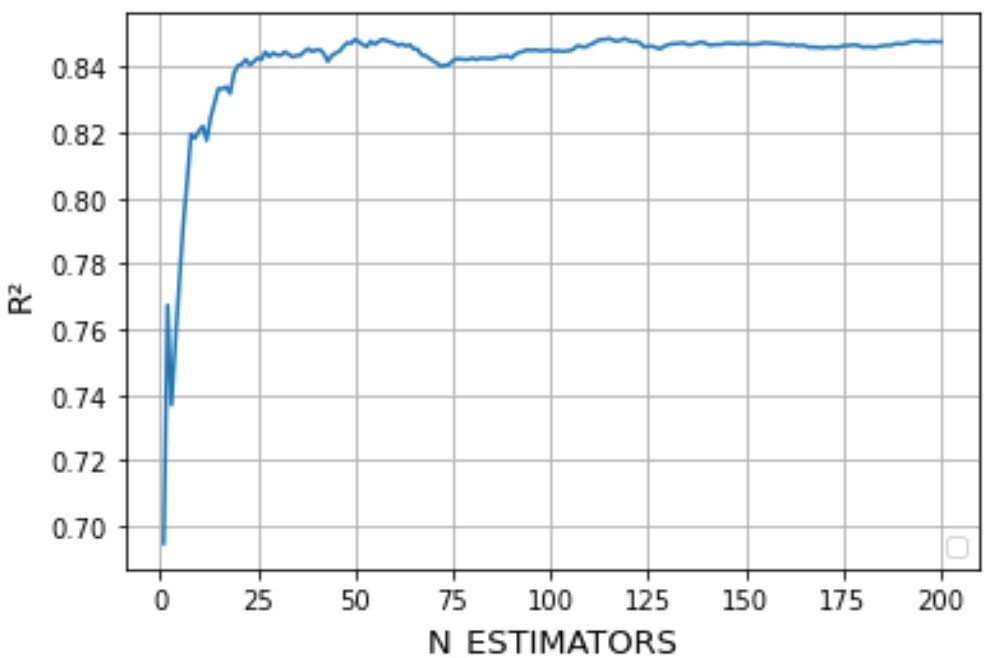

**Figure 7.** Parametric learning curve of random forest.

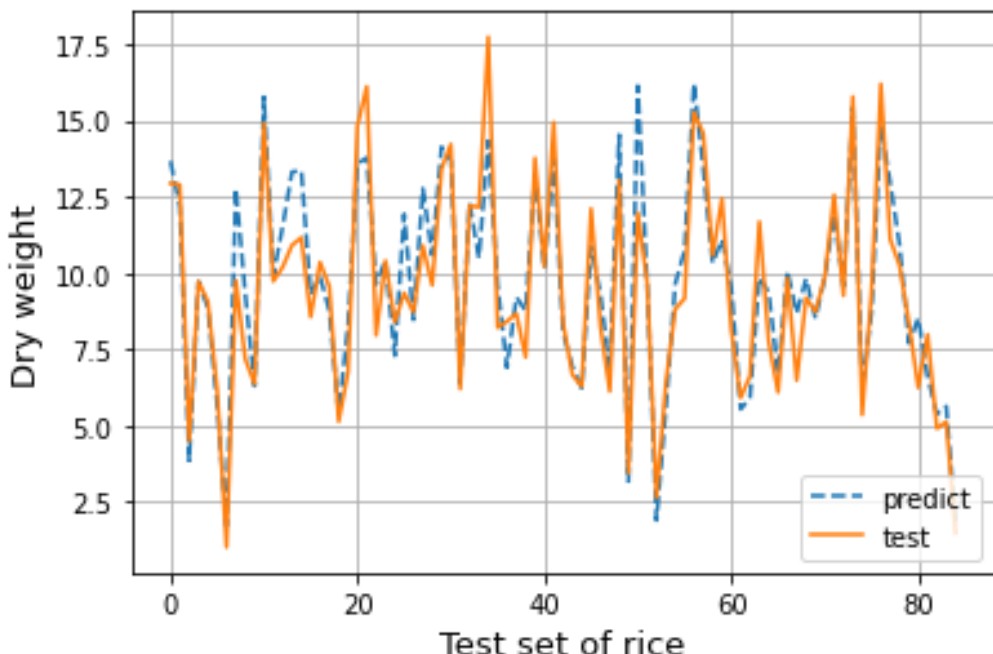

**Figure 8.** Fitting renderings of the random forest model.

In order to verify the effectiveness of the fractal dimension in predicting rice dry weight, fresh weight and plant height, the model is established by removing the fractal dimension during modeling, and the established model is compared with the model with the added fractal dimension; their $R^2$ and $MAPE$ are compared. If the fractal dimension

is valid for modeling, the $R^2$ of these models will be improved and the $MAPE$ will be decreased. The comparison results are shown in Tables 5–7.

**Table 5.** $R^2$ and $MAPE$ for the dry weight model.

| The Model Variables | $R^2$ | $MAPE$ |
|---|---|---|
| Without fractal dimension | 0.8324 | 20.67% |
| +IFD | 0.8410 | 20.66% |
| +SFD | 0.8445 | 20.63% |
| +IFD, SFD | 0.8593 | 19.66% |

**Table 6.** $R^2$ and $MAPE$ for the fresh weight model.

| The Model Variables | $R^2$ | $MAPE$ |
|---|---|---|
| Without fractal dimension | 0.8080 | 17.24% |
| +IFD | 0.8312 | 16.35% |
| +SFD | 0.8304 | 16.10% |
| +IFD, SFD | 0.8452 | 15.96% |

**Table 7.** $R^2$ and $MAPE$ for the plant height model.

| The Model Variables | $R^2$ | $MAPE$ |
|---|---|---|
| Without fractal dimension | 0.8258 | 19.85% |
| +SFD | 0.8312 | 18.93% |

It can be seen that both the fractal dimension of the rice binary image and the fractal dimension of the rectangle surrounding the rice image have a promoting effect on the fitting effect of the model. For dry weight and fresh weight models of rice, the best effect can be obtained by adding the fractal dimension of the binary image and fractal dimension of the rectangle surrounding the rice image. Although plant height has a low correlation with the fractal dimension of the binary image, it is still closely related to the fractal dimension of the rectangle surrounding the rice image.

With the growth of rice, some characters of rice change greatly, so the characteristic data change to a certain extent. By calculating the correlation coefficient of each stage, it is proved that the characteristic variables of the model did not need to change with the growth stage, but the fitting effect of dry weight, fresh weight and plant height models would all change with the growth stage. The corresponding model $R^2$ and $MAPE$ of the three stages are shown in Tables 8–10.

**Table 8.** $R^2$ and $MAPE$ for the random forest dry weight model.

| Growth Stage | $R^2$ | $MAPE$ |
|---|---|---|
| The first stage | 0.8593 | 19.66% |
| The second stage | 0.8532 | 19.98% |
| The third stage | 0.8460 | 21.85% |

**Table 9.** $R^2$ and $MAPE$ for the random forest fresh weight model.

| Growth Stage | $R^2$ | $MAPE$ |
|---|---|---|
| The first stage | 0.8452 | 15.96% |
| The second stage | 0.8421 | 16.42% |
| The third stage | 0.8218 | 18.98% |

**Table 10.** $R^2$ and $MAPE$ for the random forest plant height model.

| Growth Stage | $R^2$ | $MAPE$ |
|---|---|---|
| The first stage | 0.8312 | 18.93% |
| The second stage | 0.8428 | 18.85% |
| The third stage | 0.8889 | 20.88% |

By comparing Tables 8–10, it can be seen that when random forest is used to build the prediction model, the prediction accuracy of the model will also change with the change of the growth stage. Both the dry weight prediction model and the fresh weight prediction model show a downward trend, but even in the third stage, the $R^2$ and $MAPE$ of the model are still good. However, the plant height prediction model showed an opposite trend. With the growth of rice, the accuracy of the plant height prediction model will increase continuously. In conclusion, although the growth stage will have a certain influence on the effect of the random forest model, the influence is not significant and is within the acceptable range. Therefore, the random forest model has a good effect on the prediction of dry weight, fresh weight and plant height of rice.

### 3.2. SVR Model

In this section, the rice characteristics selected above are taken as explanatory variables. The dry weight, fresh weight and plant height are explained variables. The sklearn package of Python is used to establish prediction models of rice dry weight, fresh weight and plant height based on support vector regression. The grid search method of 10-fold cross-validation method is used to find the optimal combination of these two parameters, so that the model can achieve the optimal prediction effect. The optimization interval is determined as C belonging to 0 to 150 and Gamma to 0 to 10, respectively. A more detailed search is conducted with a step size of 1. According to the grid search results, when C equals 100 and gamma equals 4, the model achieves the highest $R^2$.

Based on the selected kernel function and the optimal parameter combination, the SVR function in the Python sklearn package is used to build the model. The 'PW', 'SA', 'SA/pH (V) × PW', 'IFD', 'SFD', 'G_g', 'f1' and 'LD1' of three growth stages are used as explanatory variables to establish a prediction model for fresh weight (g) and dry weight (g). 'PW', 'PH(V)', 'PH', 'SFD', 'G_g', 'f1', 'f2' and 'LD2' are used as explanatory variables to establish a prediction model of plant height (cm).

The data need to be normalized first, and then divided into a training set and test set in a ratio of 4 to 1. After the establishment of the SVR model, the corresponding prediction results can be obtained by substituting the data as the test set into the model. The predicted results are drawn into a line chart for fitting with the actual situation of the data in the test set, as shown in Figure 9. It can be seen from the figure that the predicted results of the model are close to the actual situation, and the distribution of data is well fitted. The prediction ability of the model is relatively stable. Under certain conditions, the predicted value of the model established in this section has certain guiding significance.

According to the above, the growth stage of rice will have an impact on the prediction performance of the model. The $R^2$ and $MAPE$ of the prediction model at different stages are compared, and the comparison results are shown in Tables 11–13.

**Table 11.** $R^2$ and $MAPE$ for the SVR dry weight model.

| Growth Stage | $R^2$ | $MAPE$ |
|---|---|---|
| The first stage | 0.8499 | 22.58% |
| The second stage | 0.6639 | 28.35% |
| The third stage | 0.7727 | 28.04% |

**Table 12.** $R^2$ and *MAPE* for the SVR fresh weight model.

| Growth Stage | $R^2$ | *MAPE* |
| --- | --- | --- |
| The first stage | 0.8032 | 23.82% |
| The second stage | 0.8152 | 25.65% |
| The third stage | 0.8551 | 24.66% |

**Table 13.** $R^2$ and *MAPE* for the SVR plant height model.

| Growth Stage | $R^2$ | *MAPE* |
| --- | --- | --- |
| The first stage | 0.8025 | 20.08% |
| The second stage | 0.8491 | 25.34% |
| The third stage | 0.8660 | 21.86% |

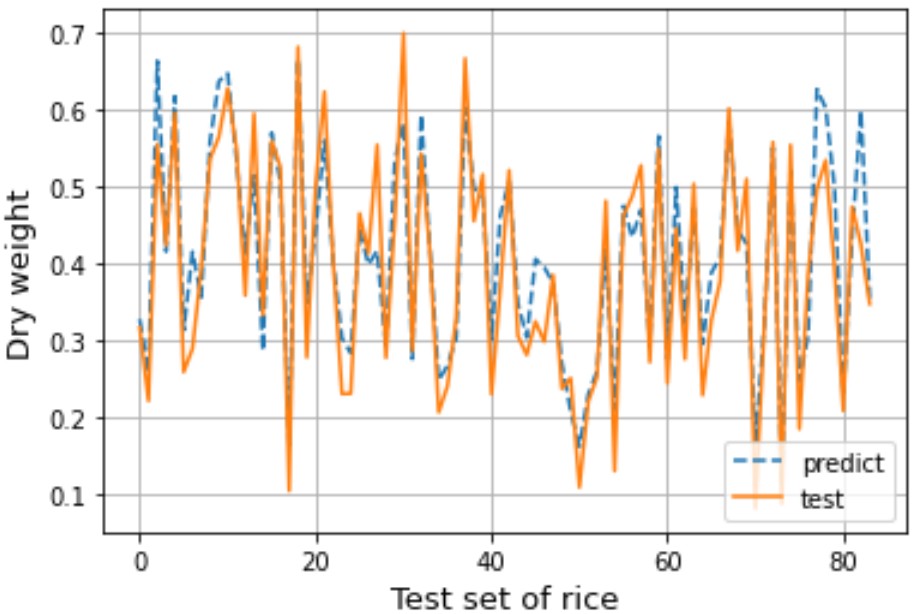

**Figure 9.** Fitting renderings of the SVR model.

By comparing Tables 11–13, it can be seen that with the growth of rice, the $R^2$ of the support vector regression model will change greatly and *MAPE* will hold steady. With the growth of rice, the prediction effect of fresh weight and plant height models will become better and better, and the increase in both models is greater than 0.05 at the third stage, which has a better prediction effect. However, with the growth of rice, the prediction effect of the dry weight model shows a downward trend; the $R^2$ of the second stage is only 0.6639, and even in the third stage, the $R^2$ of the model is less than 0.8. Although the $R^2$ of the dry weight model meets the application standard of the model, the effect is too poor compared with the model obtained by other algorithms. Therefore, support vector regression is not recommended to be used when establishing the dry weight prediction model of rice. However, support vector regression can achieve better results when establishing the prediction model of rice fresh weight and plant height.

*3.3. Linear Regression Model*

This section uses Python to build a multivariate LinearRegression prediction model using the LinearRegression function in the Sklearn package. The 'PW', 'SA', 'SA/PH (V) × PW', 'IFD', 'SFD', 'G_g', 'f1' and 'LD1' of three growth stages are used as explanatory variables to establish a prediction model for fresh weight (g) and dry weight (g). Using 'PW',

'PH(V)', 'PH', 'SFD', 'G_g', 'f1', 'f2' and 'LD2' as explanatory variables, the plant height (cm) prediction model is established.

The data are divided into a training set and test set in a ratio of 4 to 1. After the establishment of a multiple linear regression model, the corresponding prediction results can be obtained by substituting the test set data into the model. The predicted results are drawn into a line chart for fitting with the actual situation in the test set, as shown in Figure 10. It can be seen from the figure that the predicted results of the model are close to the actual situation, and the distribution of data is well fitted. The prediction ability of the model is relatively stable. Under certain conditions, the predicted value of the multiple linear regression model established in this section has certain guiding significance.

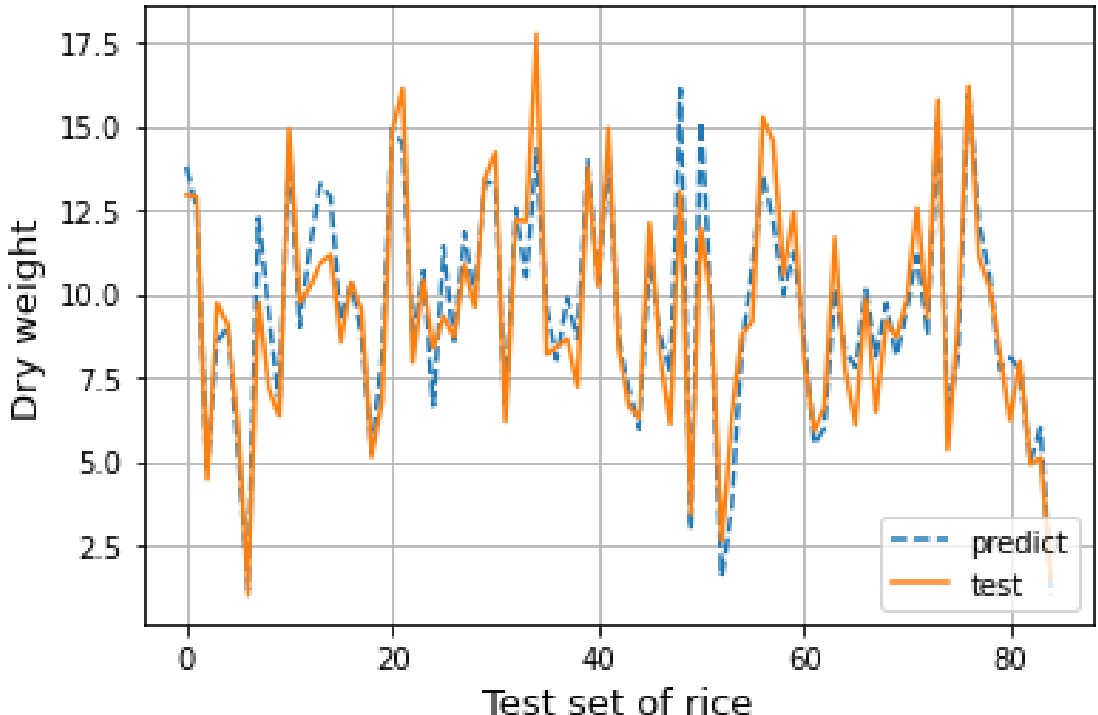

**Figure 10.** Fitting renderings of the linear regression model.

Since the growth stage of rice will affect the prediction performance of the model, the $R^2$ and $MAPE$ of multiple linear regression prediction models at different stages are compared, and the comparison results are shown in Tables 14–16.

**Table 14.** $R^2$ and $MAPE$ for the linear regression dry weight model.

| Growth Stage | $R^2$ | $MAPE$ |
| --- | --- | --- |
| The first stage | 0.8697 | 11.22% |
| The second stage | 0.8561 | 14.29% |
| The third stage | 0.8375 | 15.93% |

**Table 15.** $R^2$ and $MAPE$ for the linear regression fresh weight model.

| Growth Stage | $R^2$ | $MAPE$ |
| --- | --- | --- |
| The first stage | 0.8631 | 10.71% |
| The second stage | 0.8529 | 11.79% |
| The third stage | 0.8462 | 13.76% |

**Table 16.** $R^2$ and *MAPE* for the linear regression plant height model.

| Growth Stage | $R^2$ | *MAPE* |
|---|---|---|
| The first stage | 0.8358 | 5.27% |
| The second stage | 0.8067 | 5.55% |
| The third stage | 0.9196 | 4.45% |

By comparing Tables 14–16, it can be seen that when using the multiple linear regression model to predict the dry weight and fresh weight, the prediction effect of the model will decrease slightly with the growth of rice, but the decline is small and the overall effect is relatively stable, and good results can be achieved. However, the $R^2$ of the plant height prediction model fluctuates greatly, with the minimum value of 0.8067 in the second stage and the maximum value of 0.9196 in the third stage. The prediction effect of the third stage is the best among the models which are selected in the previous parts of this paper. By comparing the *MAPE* of all the models, the *MAPE* of linear regression models are much lower than those of random forest models and SVR models.

## 4. Discussion

In Yang Wanli's research [32], he used a SegNet convolutional neural network to segment rice images, and extracted texture features, morphological features and color features from the segmented images. He used the selected features to build a linear regression model to predict the fresh weight and dry weight of rice, and the $R^2$ of the model was 0.812 and 0.772. The characteristic parameters used in this paper are almost the same as those in Yang's paper, and the planting conditions of rice are also the same. However, fractal dimension is not paid attention to in Yang's paper, which may be the reason why the effect of Yang's model is inferior to that in this paper. Therefore, when establishing the prediction model of rice biomass, we can consider adding the fractal dimension into the model, which may improve the model effect. In Gong Hongju's research [33], she proposed the idea and method of establishing a mathematical model for accurate prediction of rice yield based on texture analysis and fractal theory. She obtained the texture features and fractal dimension by analyzing and calculating the binary image of rice, and then used principal component analysis to obtain the variables for establishing the model. She used linear regression to establish the yield prediction model, and the $R^2$ only reached 0.494, and the model had a poor effect in the posterior error test. Gong Hongju's use of the fractal dimension to establish the model is innovative, but she ignored the traditional characteristic parameters of rice, which may be the reason for the poor final result of the model. By combining the fractal dimension with the traditional characteristics of rice, the prediction model obtained in this paper has a better effect. Compared with their research, this paper improved the shortcomings of their research and obtained a simple and effective model for predicting rice biomass. The model is based on images of rice and can play a role in the field of smart agriculture.

The multivariate linear regression model is very different from other machine learning algorithms. The model obtained through machine learning belongs to the black box model, and we cannot know the specific structure of the model, nor the role of each explanatory variable in the model, so the model has a low explanatory ability. However, the linear regression model has good explanatory properties, the expression of the model can be obtained and the effect of each variable in the model can be clearly understood through the expression. In the study of rice, researchers often pay close attention to the influence of variables which can guide the improvement in varieties. Therefore, in the prediction of rice biomass, if the effect of the linear regression model is not far from that of the machine learning model, it is suggested to give priority to the linear regression model.

## 5. Conclusions

In this study, a prediction model of rice biomass based on the fractal dimension was proposed. We obtained the image of rice through the camera and extracted the phenotypic

characteristics of rice. The fractal dimension of the rice image was calculated by the box-counting method. Based on the combination of traditional characterization and the fractal dimension, prediction models were established by using random forest, SVR and multiple linear regression to predict the dry weight, fresh weight and plant height of rice. The model achieved an excellent prediction effect. We can draw the following conclusions.

(a)  As an image feature of rice, the fractal dimension can be used to predict the biomass of rice. According to the image and correlation coefficient, the fractal dimension is correlated with dry weight, fresh weight and plant height of rice. The fractal dimension can improve the effect of the biomass prediction model. Therefore, the fractal dimension can be considered to achieve better results in future research on rice.

(b)  By using various methods to establish the biomass prediction model of rice, it can be found that the effect of the linear regression model is better than random forest and SVR. Compared with the random forest model and the SVR model, the multiple linear regression model has the largest $R^2$ and the smallest *MAPE*, and has a good explainability. Therefore, when using rice images to build models to predict dry weight, fresh weight and plant height, multiple linear regression models are preferred. In addition, random forest and SVR are not recommended.

(c)  Although the growth stage of rice will affect the prediction effect of the model, the fluctuation is within the acceptable range. Compared with the models established above, only the SVR dry weight model shows a great decline in $R^2$. The $R^2$ of the other models can remain above 0.8. Although the stage of rice growth has changed, the *MAPE* of these models remain stable. The overall effects of these models are still quite excellent.

**Author Contributions:** Conceptualization, Y.H. and J.S.; methodology, Y.H.; software, Y.H.; validation, J.S. and Y.H.; formal analysis, Y.H.; investigation, Y.H.; resources, J.S. and Y.Q.; data curation, J.S.; writing—original draft preparation, ALL; writing—review and editing, Y.H. and J.S.; visualization, Y.H.; supervision, J.S.; project administration, J.S.; funding acquisition, J.S. and Y.Q. All authors have read and agreed to the published version of the manuscript.

**Funding:** This study was supported by the National Strategic Science and Technology Development Fund (Grant No. G20200017074), National High-End Foreign Expert Introduction Program, Master Studio of Huazhong Agricultural University, National Natural Science Foundation of China (Grant No. 31701317), Fundamental Research Funds for the Central Universities (Grant No. 2662020GXPY007), Science and Technology Innovation Fund of Huazhong Agricultural University (Grant No. 2021367) and National Innovation and Entrepreneurship Training Program for College Students (Grant No. S202010504228).

**Data Availability Statement:** Collected data are available from the authors. The codes are stored in https://github.com/huyijunhenshuai/rice-prediction.git access on 17 June 2021.

**Acknowledgments:** The authors would like to thank the three anonymous referees for their constructive comments and suggestions. Thanks to Lingfeng Duan for her support and guidance in this research.

**Conflicts of Interest:** The authors declare no conflict of interest.

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
