# Peer review of "Estimation of Rice Biomass at Different Growth Stages by Using Fractal Dimension in Image Processing"

_applsci, doi:10.3390/app11157151_

Round 1
Reviewer 1 Report
A new parameter – fractal dimension utilized in the process of rice plant growth prediction, was proposed in the study. The proposed parameter was used in three estimation methods – random forest model, SVR, and linear regression.
I have some advice that should be incorporated in the text before study publishing:
- Parameter N (r. 198) should be replaced because it is used in different meanings (r. 186). Or is it the same parameter?
- All rice parameters in Tab. 1 should be well described in the text.
- Why are the dry weight and fresh weight analyzed? Isn’t the difference only in water? Is the dry and fresh weight evaluated for the same set of plants?
- Were the learning process and the evaluation process applied to the same test plants?
- Figure. 4 has probably the wrong title.
- Correlation description (r. 230-234) is redundant especially when the other important parameters are not well described.
- What are LD1 and LD2?
- How the Texture feature (G_g) was determined?
- Although the proposed parameter was used in three estimation methods, it would be good if the proposed method were evaluated with other state-of-the-art methods.
- For a better understanding of the proposed parameter fractal dimension, it should be better described how it is determined and it should be described with images.
- In my opinion, the methods and materials should be more deeply described.
Reviewer 2 Report
The article is interesting, well presented, it is possible to publish
Reviewer 3 Report
The reviewed article deals with the fractal dimension for smart agriculture that can improve the performance of the prediction model. One of the important things about the fractal dimension that could be calculated from the image, and here authors used for rice. The authors found correlation coefficients with a strong correlation with biomass. The paper is generally well structured and well written overall but I some general comments.
- There are many grammatical errors and needs to check also several spell/typo errors
- How many replications were done for the binary image analysis? It is not mentioned.
- In conclusion, parts must need to improve as it is described in general but not any findings. Here should give more priority on writing to the overall conclusion.
- Authors which model gave more priority for another crop analysis besides linear regression model it is not clear.
Round 2
Reviewer 1 Report
The revisited version fulfills my comments and suggestions.
Author Response
Dear reviewer,
Thank you again for your questions, and we sincerely appreciate your comments. Your comment is of instructive significance, which makes our manuscript more perfect. We have revised the title of the article in response to the editor's comments and hope that the article will be published successfully. We are committed to solving a scientific problem and hope that our manuscript can make a relatively large contribution to the research of rice.
In the end, we deeply appreciate your again for your comments!
Thank you and best regards!
Yours sincerely,
Yijun Hu
Reviewer 3 Report
The revised manuscript by Hu et al deals with an efficient prediction
3 model based on fractal dimension in rice for smart agriculture. The Article was critically revised and corrected by the authors according to the Reviewers comments and directions. Thus, The present format of this manuscript could be considered for full publication in the Journal of Applied Sciences.
Author Response

(The authors gave the same response as above.)
